# Analysis of the coaptation role of the deltoid in reverse shoulder arthroplasty. A preliminary biomechanical study

Lucas Martinez[1]☉*, Margaux Machefert[1]☉, Thomas Poirier[1]☉, Jean Matsoukis[2]☉, Fabien Billuart[1,3]☉

**1** Laboratoire d'Analyse du Mouvement, Institut de Formation en Masso-Kinésithérapie Saint Michel, Paris, France, **2** Département de chirurgie orthopédique, Groupe Hospitalier du Havre, Le Havre, France, **3** Unité de Recherche ERPHAN, UR 20201, Université de Versailles Saint Quentin, Versailles, France

☉ These authors contributed equally to this work.
* l.martinez@ifmk.fr

## Abstract

### Background

Lateralization of the glenoid implant improves functional outcomes in Reverse Shoulder Arthroplasty. Lateralization does not appear to impact the Deltoid's Moment Arm. Therefore, the stabilizing effect described in the literature would not be the result of an increase this moment arm. A static biomechanical model, derived from Magnetic Resonance Imaging, can be used to assess the coaptation effect of the Middle Deltoid. The objective of this study was to analyze the impact of increasing amounts of glenoid lateralization on the moment arm but also on its coaptation effect.

### Methods

Eight patients (72.6 ± 6.5 years) operated for Reverse Shoulder Arthroplasty were included in the study. Three-dimensional models of each shoulder were created based on imaging taken at 6 months postoperative. A least square sphere representing the prosthetic implant was added to each 3D models. A static biomechanical model was then applied to different planar portions of the Middle Deltoid (from 3D models), first without lateralization and then with simulated lateralization of 6, 9 and 12mm. This static model enables to compute a Coaptation/Elevation Ratio and to measure the Deltoid's Moment Arm. The inter- and intra-rater agreement of the 3D models was evaluated.

### Results

One patient was excluded due to motion during imaging. The inter- and intra-rater agreement was over 0.99. The ratio increased starting at 6 mm of lateralization ($p<0.05$), compared to the initial position. The moment arm was not affected by lateralization ($p<0.05$), except in two slices starting at 9 mm (S1 $p<0.05$ and S2 $p<0.05$).

**Data Availability Statement:** We uploaded the data to Figshare. Here is the DOI: 10.6084/m9.figshare.15066207.

**Funding:** The authors received no specific funding for this work.

**Competing interests:** The authors have declared that no competing interests exist.

## Conclusion

Our hypothesis that the Middle Deltoid's coaptation role would be greater with glenosphere lateralization was confirmed. This trend was not found in the moment arm, which showed little sensitivity to lateralization. The stabilizing effect therefore appears to stem from the coaptation role of the Middle Deltoid.

## 1. Introduction

The Reverse Shoulder Arthroplasty (RSA) technique developed by Grammont aimed to lower and medialize the glenohumeral joint's center of rotation and increase the Deltoid's Moment Arm (DMA) [1–3]. Scapular notching remains a common complication of RSA [1–12]. Hettrich et al. [4] define scapular notching as the result of a mechanical conflict between the medial part of the metaglene against the lateral edge of the scapula during adduction movements. Notching is therefore responsible for progressive osteolysis of the glenoid as well as premature wear of the metaglene, making the implant unstable. This is the major complication of this type of arthroplasty. According to Rugg et al. [3], scapular notching is thought to be present in nearly 2/3 of RSA 2 years postoperatively. Hettrich et al. [4] even estimate that notching occurs in 31% to 97% of patients undergoing RSA. Clinically, notching is evidenced by loss of muscle strength, pain, deterioration of active mobility of the shoulder in flexion and abduction [3]. Hoenecke et al. [13] believe that optimal implant positioning is a trade-off between this potential complication and deltoid efficacy. Lateralization can help minimize medialization, which causes notching, using either the glenoid or humeral implant, or a glenoid bone graft [11, 14, 15]. For Boileau et al. [8–12], lateralization is a key element to achieve adequate passive range of motion while minimizing the risk of notching. Tightening of the deltoid through lateralization would help increase joint stability [10–12]. Although lateralization is widely studied, the ideal amount required to improve shoulder mobility, stability and strength—while simultaneously decreasing notching—has yet to be determined. Studies report lateralization between 1 mm and 13 mm [1–13]. For Boileau et al., graft thickness should be 10 mm [9, 11, 16]. However, lateralization does not seem to impact the DMA [14]. Therefore, it is impossible to explain the stabilizing effect described by Boileau et al. [10–12] by changes in this DMA. Li et al. [17] and Smith et al. [18] showed that the Middle Deltoid (MD) has key role after RSA with an increase of EMG activity after surgery. Thus, a specific focus on this part of the deltoid muscle is clinically relevant.

A more in-depth mechanical study of the MD will help define its stabilizing role. Gagey et al. [19], Billuart et al. [20] and Hereter Gregory et al. [21] have developed a model based on Magnetic Resonance Imaging (MRI) studies to characterize the coaptation and elevator role of each portion of the MD using a Coaptation/Elevation Ratio (CER). This ratio is computed between the "elevating" force (which raises the humeral head and "destabilizes" the shoulder) and "lowering" force (which stabilizes the humeral head) using angle measurements and force estimation. This ratio thus makes it possible to characterize the main action of the middle deltoid on the glenohumeral joint in a static and plane condition. This model, applied to patients suffering from shoulder osteoarthritis, showed that the MD played an important role as an elevator muscle but that it had a coaptation component as well, stabilizing the shoulder [19].

Therefore, the primary objective of this preliminary study was to analyze the impact of an increase in glenosphere lateralization on the DMA, but also on the coaptation role of MD. By comparing the initial position of the glenosphere with simulated lateralizations, our hypothesis

is that lateralization will increase the coaptation role of the MD without impacting its DMA. Assessing the reliability of the 3D models was the study's secondary objective.

## 2. Materials & methods

The Ethics Committee of the Groupe Hospitalier du Havre (GHH) approved and deemed the protocol of this study to comply with the ethics rules on clinical research on January 15[th], 2021. Experimental work including human subjects within the framework of this study can be implemented. Each patient was informed of the study via a newsletter. Then, each patient gave their consent to participate in the study by signing a consent letter informing them that their data will be used for the current study. Patient data has been anonymized.

### 2.1 Patients

Eight RSA patients were included at the beginning of the study, from March 8[th] to March 12[th] 2021 (Table 1). The inclusion criteria were RSA performed less than 6 months prior and an indication of cuff tear arthropathy diagnosed with imaging. Exclusion criteria were a history of surgery or trauma to the affected shoulder, the presence of complications, such as dislocation or surgical site infection and other severe medical conditions (neurological, cardiovascular, oncological). The exclusion criteria were screened for on MRI. Finally, movements of the patient (voluntary or not) during the MRI examination, rendering the images uninterpretable was the last exclusion criterion (thus one patient was excluded after imaging). All the patients were treated by the same surgeon between February 1[st] 2021 and February 5[th] 2021, using the deltoid-splitting approach without lateralization of the glenoid implant.

### 2.2 MRI protocol

The images were taken with a 1.5T MRI scanner (Siemens™ Magnetom Aera, Munich, Germany). The subjects were supine, with an angle of 80˚ between the humeral diaphysis and the scapula (measured with a goniometer) and palms up. 3D measurement of this angle was not possible due to artefacts on the MRI near the glenosphere. The images were taken in Spin-Echo T1-weighted sequence (472/8 for the coronal plane and 350/7.5 for the axial plane, field of view: 560X640) with a fat hypersignal suppression option: Fat Sat 1. Slice thickness was 3 mm, joined without overlapping.

### 2.3 3D modeling

The deltoid, the humeral diaphysis, the clavicle and acromion were manually segmented on each slice and automatically reconstructed in 3D with the SliceOmatic™ software *(Tomovision™,*

**Table 1. Patient characteristics.**

| Patients | Sex | Age | OS | Implant used | Glenosphere (mm) |
|---|---|---|---|---|---|
| *P1* | F | 67 | L | *Aequalis Ascend Flex*, **Wright-Tornier™** | 36 |
| *P2* | F | 67 | R | *Delta XTEND*, **DePuy-Synthes™** | 38 |
| P3 | M | 78 | R | *Delta XTEND*, **DePuy-Synthes™** | 38 |
| *P4* | F | 79 | R | *Delta XTEND*, **DePuy-Synthes™** | 38 |
| P5 | F | 82 | L | *Delta XTEND*, **DePuy-Synthes™** | 38 |
| *P6* | M | 67 | R | *Delta XTEND*, **DePuy-Synthes™** | 38 |
| P7 | F | 66 | L | *Aequalis Ascend Flex* **Wright-Tornier™** | 36 |
| *P8* | F | 75 | L | *Delta XTEND*, **DePuy-Synthes™** | 38 |

P = Patients; OS = Operated Shoulder; F = Female; M = Male; L = Left; R = Right

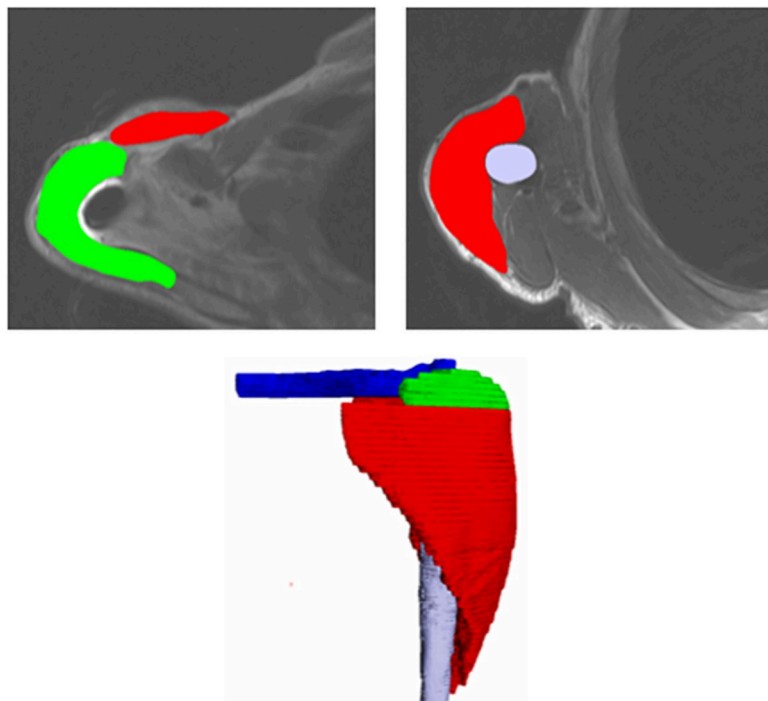

**Fig 1. Manual segmentation of the slices and 3D modeling with SliceOmatic™ software.** In Green: acromion; in Red: deltoid; in Blue: clavicle; in Grey: humeral diaphysis.

*Montreal, Canada).* The implants could not be reconstructed because of the presence of several artifacts (Fig 1).

## 2.4 Static analysis

The model used in this study computed the CER on MD planar slices, taken from the 3D models. The MD was defined as a mechanical system and the external forces applied to it were inventoried (Fig 2). Mass and thickness of MD, and other muscles were not taken into consideration. The MD encapsulates the implant like a string on a pulley. The MD's action on the pulley at its distal insertion site is defined as $\overrightarrow{F'_1}$ with $\overrightarrow{F'_1} = \overrightarrow{-F_1}$. The action of MD on the pulley is defined as $\overrightarrow{R\prime}$ with $\overrightarrow{R\prime} = \overrightarrow{-R}$. Considering the external forces inventoried and the principles of statics, the following equation was defined within the plane:

$\overrightarrow{F_1} + \overrightarrow{F_2} + \overrightarrow{R} = \overrightarrow{0}$.

The forces projected on the y'y axis were as follows:

$Ry = F1 \cos(E) + F1 \cos(2T+E)$

Hence: $Ry = 2F1 \cos(T) \cos(T+E)$

The CER is computed based on standard forces directly in contact with the humerus:

- Coaptation force = action of MD on the pulley, as $R'y$

- Elevator force = distal effect of the MD on the humerus, as: $F'_1y$

  The formula used was: $CER = \frac{R'y}{F'_1y}$.

  Finally:

- $R'y = -2\,F_1\cos(T)\cos(T+E)$

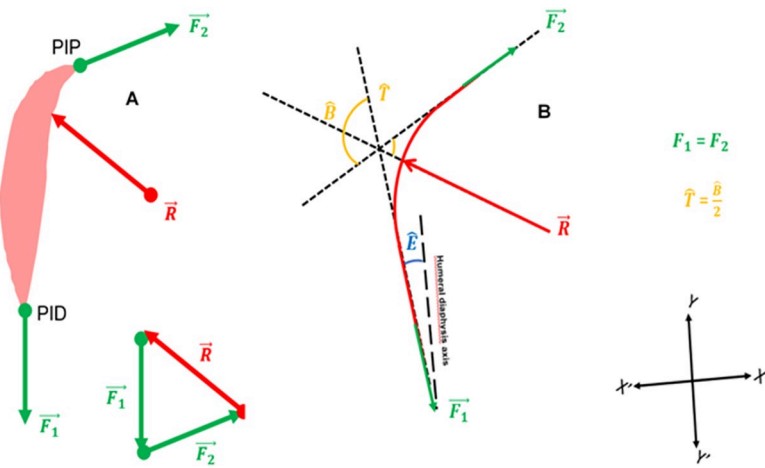

**Fig 2. Biomechanical models used in the present study.** (A) External forces and fundamental principles of the static model. $\overrightarrow{F_1}$ corresponds to the distal insertion Point (PID). $\overrightarrow{F_2}$ is the proximal insertion point (PIP). $\overrightarrow{F_1} = \overrightarrow{F_2}$. $\overrightarrow{R}$ is equivalent to the reaction force on the MD. (B) The Y'Y axis is parallel to the humeral diaphysis. Ê represents the angle between MD muscle fiber orientation and the humeral diaphysis axis. B̂ represents the MD's change in direction. T̂ = B̂/2. Only the normal $F1$ and $R$ vectors are used to compute the CER (forces directly in contact with the humerus). This model makes it possible to compute the CER using angle measurements.

- $F'_1y = F_1 \cos(E)$

$$\frac{R'y}{F'1y} = \frac{-2F_1\cos(T)\cos(T+E)}{F_1\cos(E)} = \frac{-2\cos(T)\cos(T+E)}{\cos(E)}$$

Finally, to compute the CER, angle $\hat{T}$ and $\hat{E}$ have to be measured on MD planar slices.

## 2.5 3D model segmentation

To compute the CER based on $\hat{T}$ and $\hat{E}$, we used various MD slices of the planar model extracted from the 3D models (Fig 2B). For this part of the study the mechanical model developed and validated by Billuart *et al.* [20] and then by Hereter Gregori *et al.* [21] on degenerative shoulders was used and transposed to prosthetic shoulders. In the present study, the following methodology was used: the MD was considered the sum of "n" muscle fibers, starting at the anterior and lateral aspect of the acromion and terminating on the deltoid V (DIP) which lay on a sphere representing the contact area between the prosthetic implant and the internal aspect of the MD. This sphere called ILSS (Implant Least Squared Sphere) was positioned within the 3D model using the least squared method in SliceOmatic™. The following methodology was used to build ILSS: a mesh sphere was created in the software, which allows manual point selection on the internal aspect of the MD. Its size (radius ILSS) and its position were adjusted manually in the 3D model, as close to the desired position as possible (Fig 3). Then the least square methodology was used: the software eliminates all the points for which the normal vectors are not in the same direction as the sphere's normal vector. If the scalar product between the two directions: "normal" x "dir (node—center)" is over 0, the point is eliminated. Then, the software's "distance" function eliminates 10% (with each click) of the points most distant from the surface of the sphere, using the following equation: $dist^2 = ABS$ ((node-cent)2 $-r^2$)). The radius of this sphere was identified as ILSS radius and recorded.

Then, the 3D models were segmented in several plane sections using 3 points:

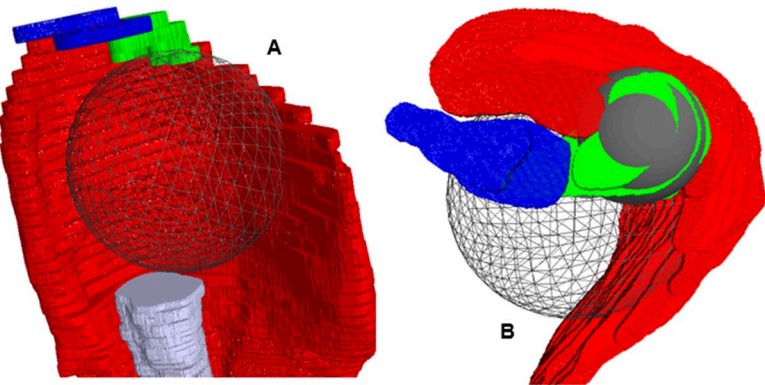

**Fig 3. Positioning of the least square spheres (ILSS and acromion). (A)** A mesh sphere is created which allows manual point selection on the internal aspect of the MD: its size (radius ILSS) and position are adjusted manually in the 3D model, as close to the desired position as possible. **(B)** The grey sphere is the 2nd least squared sphere going through the acromion to find Point 1.

- *Point 1*: center of a 2nd least squared sphere going through manually chosen points on the acromion, using the same methodology as ILSS (Fig 3B).

- *Point 2*: manual selection on A2 insertion of MD [22], changing in 10˚ increments with each slice and running through all the acromion (Fig 4).

- *Point 3*: center of the humeral diaphysis.

From these three points, a 4x4 homogeneous transformation matrix of the new plane was defined. For each pixel of this new plane, the cubic interpolation data was computed from the eight neighboring pixels from the original model using the following formula: $2t^3 - 3t^2 + 1$,

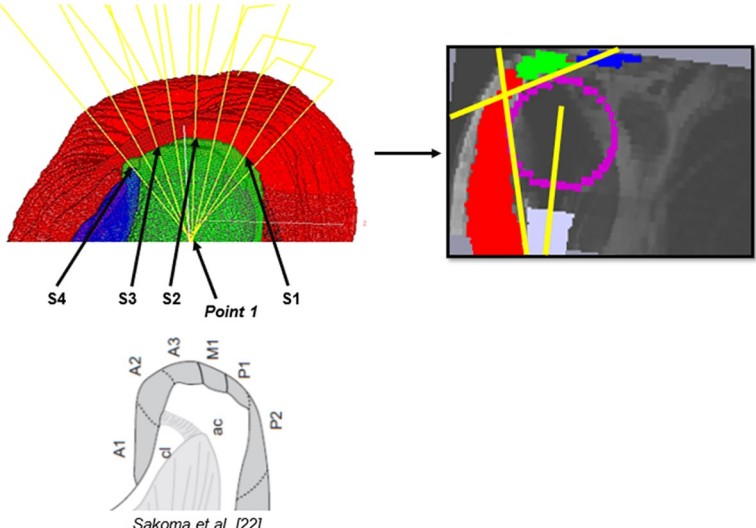

**Fig 4. Plane Slices selection and angle measurements. (A)** Point 2 changes on each slice: S4 = A2, S3 = A3, S2 = M1 and S4 = P1. **(B)** Each plane slice includes a MD string (red), a pulley (part of the ILSS, purple), a part of the acromion (green) and of the clavicle (blue). The CER is computed using the $\hat{\mathbf{T}}$ and $\hat{\mathbf{E}}$ measurements with the straight yellow lines. **(C)** Caption credit: Sakoma *et al.* [22].

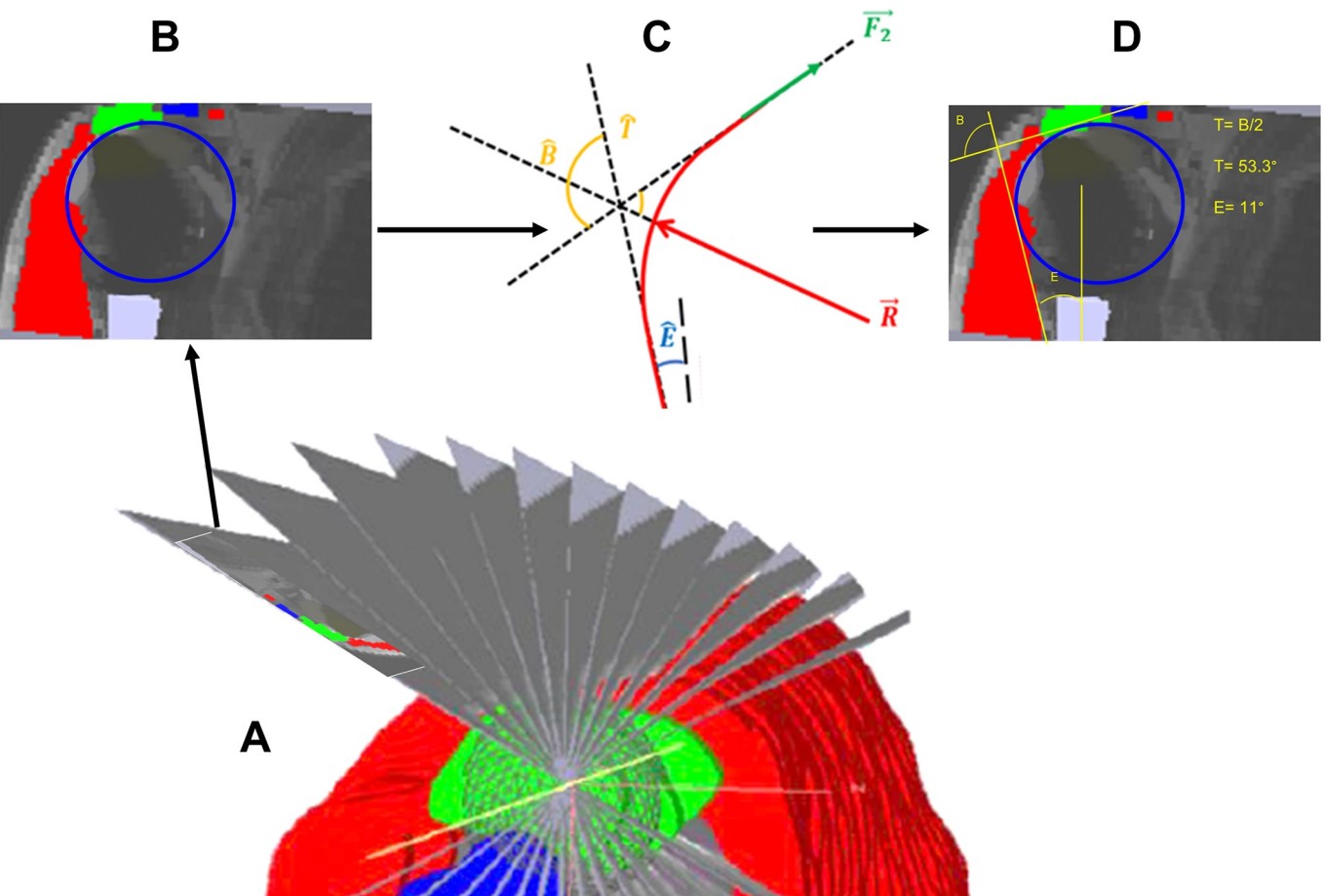

**Fig 5. Summary of the steps from the selection of slices to angle measurements. (A)** The 3D model is segmented in several plane sections according to the methodology presented above. **(B)** After selecting the slices according to the anatomical description by Sakoma *et al.* [22] and in which a portion of the ILSS is "stamped", the biomechanical model developed by Billuart *et al.* [20] **(C)** is used to carry out the angle measurements **(D)**.

where *t* is the distance between the new point and one of the points from the original model. Then, four plane slices were chosen manually for each shoulder, according to the study by Sakoma *et al.* [22]. Indeed, the slices passing through the insertion points on the acromion and corresponding to the four MD intramuscular tendons insertions described by Sakoma *et al.* [22] were selected (Fig 4C). Thus, the Slice 4 in the present study (S4) goes through the A2 insertion described by Sakoma *et al.* [22]. In the same way: S3 = A3, S2 = M1, S1 = P1 (Fig 4). The different steps from the manual selection of the slices to the incorporation of the bio-mechanical model by Billuart *et al.* [20] and then the measurement of the angles is summarized in Fig 5.

## 2.6 CER computation

The angles were measured manually on each of the four slices (Fig 4B). A straight line in contact with the pulley and following the internal aspect of the MD was called the *MD line*. The intersection between the *MD line* and the humeral diaphysis axis defines $\hat{E}$. The PIP of the MD and a point of contact with the pulley define another straight line called the *proximal line*. The intersection between the *proximal line* and the *MD line* formed $\hat{B}$, knowing that $\hat{T} = \hat{B}/2$. The ratio was then computed with the equation presented previously.

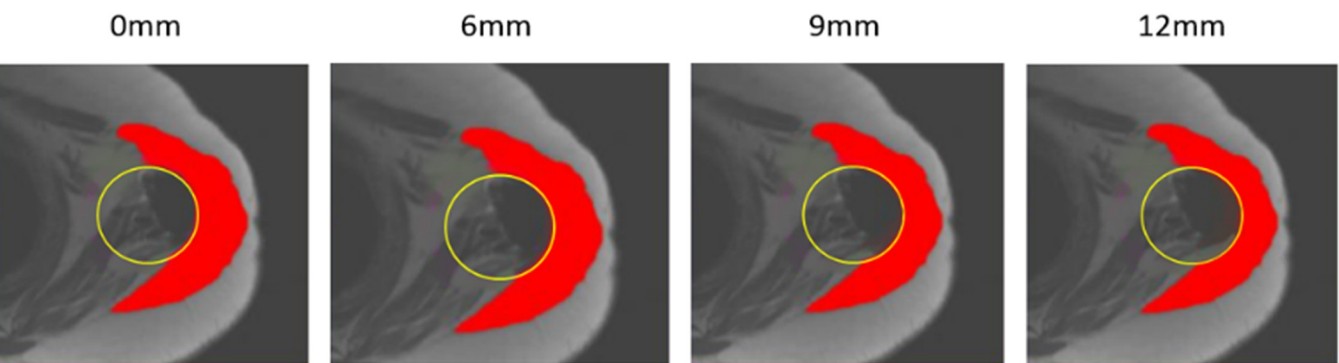

**Fig 6. Simulation of lateralizations on SliceOmatic™ in axial view.** 2D representation on an MRI image of the displacement of a 2D slice of ILSS (stamped in the MRI image) along the X axis at 0, 6, 9 and 12mm (*from left to right*). Yellow circle = slice of ILSS in 2D; In red = deltoid muscle.

### 2.7 Lateralization of the glenosphere

The lateralization was simulated in SliceOmatic™ by translating the ILSS (on the X-axis) in 6, 9 and 12 mm increments compared to its initial postoperative position without lateralization (Fig 6). For this part of the study, the following simplifying hypothesis was used: the ILSS radius was non-significantly modified by the lateralization of the prosthetic implant.

### 2.8 DMA measurement

The DMA was measured on each slice as the distance between the center of the ILSS and the line perpendicular to $\overrightarrow{F_1}$ (with the tool *"Measurement"* of the software).

### 2.9 Inter- and intra-rater assessment

To assess the reliability of the 3D models and the segmentation method, an inter- and intra-rater agreement study was undertaken. Two raters created the 3D models and positioned the ILSS. Rater 1 did the models and positioned the ILSS twice on each study subject's images, one week apart. Rater 2 completed the same protocol on all subjects. To assess intra-rater agreement, the models created one week apart were compared for each rater. For the inter-rater agreement, the models of the three common subjects were compared.

### 2.10 Statistical analysis

The data were analyzed with R version 2.14 (Bell Laboratories, Murray Hill, USA). Non-parametric tests were used because normal data distribution could not be guaranteed. The significance level was set at 0.05. The inter- and intra-rater agreement was evaluated with an intraclass correlation coefficient (ICC) for the 3D models and the ILSS radius. The mean/standard deviation of the angles, the CER and DMA were used for the analysis. The comparison of CER, angles and DMA between initial and lateralized conditions were done with the Mann-Whitney test.

## 3 Results

P2 was excluded because of motion during MRI acquisition and thus, the different MRI slices were not usable for the 3D reconstruction.

**Table 2.  Intra- and inter-rater agreement.**

|  | P1 | P3 | P4 | P5 | P6 | P7 | P8 | *Mean* |
|---|---|---|---|---|---|---|---|---|
| ICC 3D model for Rater 1 (Week 1 vs Week 2) | 0.999 | 0.999 | 0.999 | 0.999 | 0.993 | 0.999 | 0.999 | *0.999* |
| ICC 3D model for Rater 2 (Week 1 vs Week 2) | 0.998 | 0,990 | 0,990 | 0,993 | 0,999 | 0.999 | 0.999 | *0.999* |
| ICC ILSS radius for Rater 1 (Week 1 vs Week 2) | 0.997 | 0.999 | 0.998 | 0.992 | 0.997 | 0.999 | 0.990 | *0.996* |
| ICC ILSS radius for Rater 2 (Week 1 vs Week 2) | 0.995 | 0.949 | 0.992 | 0.995 | 0.907 | 0.998 | 0.999 | *0.973* |
| ICC Rater 1 vs 2: Week 1 (for 3D models) | 0.990 | 0.998 | 0.990 | 0.999 | 0.993 | 0.999 | 0.990 | *0.994* |
| ICC Rater 1 vs 2: Week 2 (for 3D models) | 0.989 | 0.997 | 0.987 | 0.999 | 0.988 | 0.991 | 0.992 | *0.989* |
| ICC Rater 1 vs 2: Week 1 (for ILSS radius) | 0.957 | 0.933 | 0.762 | 0.911 | 0.895 | 0.713 | 0.862 | *0.957* |
| ICC Rater 1 vs 2: Week 2 (for ILSS radius) | 0.957 | 0.963 | 0.877 | 0.996 | 0.929 | 0.83 | 0.925 | *0.925* |

P = Patient

## 3.1 Reliability assessment

The results from the reliability assessment are presented in Table 2. For intra-rater agreement of 3D model, the mean ICC was 0.999 for Rater 1 and Rater 2. For intra-rater agreement of ILSS radius, the mean ICC was 0.996 for Rater 1 and 0.973 for Rater 2. For inter-rater agreement of 3D models, the mean ICC was 0.994 at Week 1 and 0.989 for the second comparison at Week 2. For inter-rater agreement of ILSS radius, the mean ICC was 0.957 at Week 1 and 0.925 at Week 2.

## 3.2 ILSS radius

The ILSS radius of every patient is presented in Table 3. The mean ILSS radius was 32.89±4.29 mm.

## 3.3 Biomechanical parameters

Table 4 presents the values of the biomechanical parameters for different lateralization options and for each slice selected. *For 0 mm of lateralization*, Mean and Standard Deviation (Mean ±SD, for all slices) are 54.93±0.77˚ for Angle T; 7.94±1.24˚ for Angle E; 0.54±0.03 for the CER, and 38.44±8.80 mm for the DMA. *For 6 mm of lateralization*, Mean and Standard Deviation (Mean±SD, for all slices) are 45.60±0.99˚ for Angle T; 11.47±0.81˚ for Angle E; 0.81±0.03 for the CER, and 39.67±8.25 mm for the DMA. *For 9 mm of lateralization*, Mean and Standard Deviation (Mean±SD, for all slices) are 42.54±1.03˚ for Angle T; 14.07±0.93˚ for Angle E; 0.83 ±0.04˚ for the CER, and 40.67±9.05 mm for the DMA. *For 12 mm of lateralization*, Mean and Standard Deviation (Mean±SD, for all slices) are 40.07±0.90˚ for Angle T; 15.25±1.02˚ for Angle E; 0.88±0.03 for the CER, and 41.28±5.25 mm for the DMA.

**Table 3.  Radius of the ILSS.**

| Patients | Radius (mm) |
|---|---|
| P1 | 31.25 |
| P3 | 36.30 |
| P4 | 33.64 |
| P5 | 25.72 |
| P6 | 39.42 |
| P7 | 32.16 |
| P8 | 31.75 |
| **Mean ±SD** | **32.89 ± 4.29** |

**Table 4. Means, standard deviation, minimum and maximum values of the biomechanical parameters for different lateralization options.**

| | | 0 mm | 6 mm | 9 mm | 12 mm |
|---|---|---|---|---|---|
| ANGLE T (degrees) | S1 | 55.40±1.68 | 42.71±2.04 | 41.40±1.34 | 38.75±1.71 |
| | | (53–57) | (42–43.5) | (40–44.5) | (37–41) |
| | S2 | 55.43±2.23 | 42.86±1.86 | 41.75±1.77 | 40.20±0.45 |
| | | (52–59) | (40.5–45) | (39.5–43) | (40–36.5) |
| | S3 | 54.36±1.65 | 44.79±3.49 | 43.71±3.68 | 40.60±2.07 |
| | | (53–58) | (40–51) | (42–50) | (37–47.5) |
| | S4 | 54.14±1.31 | 44.00±2.90 | 42.50±2.78 | 40.71±1.89 |
| | | (53–56) | (40–47.5) | (39–46) | (38–43) |
| ANGLE E (degrees) | S1 | 9.14±3.44 | 13.43±2.57 | 15.14±2.61 | 16.86±2.41 |
| | | (3–13) | (10–16) | (12–19) | (14–21) |
| | S2 | 6.71±2.21 | 11.57±3.10 | 13.14±2.48 | 14.86±2.61 |
| | | (3–10) | (7–15) | (10–16) | (11–18) |
| | S3 | 7.29±1.60 | 12.29±2.43 | 13.71±2.43 | 14.57±2.15 |
| | | (4–9) | (8–16) | (9–17) | (11–17) |
| | S4 | 7.43±2.30 | 13.00±3.92 | 14.71±3.73 | 16.00±3.21 |
| | | (3–9) | (6–18) | (8–20) | (10–20) |
| CER | S1 | 0.49±0.08 | 0.82±0.08 | 0.84±0.08 | 0.90±0.05 |
| | | (0.39–0.60) | (0.71–0.97) | (0.73–0.98) | (0.86–0.98) |
| | S2 | 0.53±0.09 | 0.87±0.07 | 0.88±0.06 | 0.91±0.04 |
| | | (0.44–0.70) | (0.73–0.98) | (0.79–0.99) | (0.85–0.96) |
| | S3 | 0.55±0.06 | 0.79±0.11 | 0.79±0.11 | 0.84±0.09 |
| | | (0.44–0.62) | (0.63–0.93) | (0.67–0.98) | (0.8–1.01) |
| | S4 | 0.56±0.05 | 0.79±0.06 | 0.8±0.05 | 0.86±0.05 |
| | | (0.47–0.61) | (0.69–0.88) | (0.73–0.88) | (0.78–0.94) |
| DMA (mm) | S1 | 41.52±3.31 | 43.32±3.54 | 47.16±2.93 | 49.49±2.94 |
| | | (36.55–47.08) | (39.03–49.95) | (43.65–52.94) | (45.77–55.29) |
| | S2 | 48.65±4.45 | 50.66±4.50 | 54.31±5.07 | 56.04±5.77 |
| | | (39.71–53.17) | (48.90–56.12) | (45.29–60.34) | (45.29–63.14) |
| | S3 | 33.88±5.86 | 36.01±5.81 | 38.31±6.36 | 40.17±5.58 |
| | | (27.91–42.62) | (28.67–43.25) | (31.34–47.1) | (34.53–48.85) |
| | S4 | 28.54±6.36 | 31.09±6.33 | 33.01±6.65 | 34.43±6.59 |
| | | (21.07–38.19) | (24.05–40.01) | (25.81–42.09) | (25.99–44.03) |

S = MD slice number.

Fig 7 represents a set of images containing all slices at all lateralizations with the angles overlay for Patient 1.

## 3.4 Comparisons of lateralization simulations

The comparison of the different lateralization simulations is presented in Table 5. For Angle T and E, there are significant differences for all the slices when comparing 0mm of lateralization versus 6mm, 9mm and 12mm (significance level was p = 0.002* for Angle T and between p = 0.04* and p = 0.002* for Angle E). When lateralization increases, for all the slices, Angle T decreases and inversely, Angle E increases. CER increase significantly (significance level was between p = 0.002* and p = 0.0006*) when comparing the initial position without lateralization (0mm) versus 6mm, 9mm and 12mm, for all the slices. There was no significant difference

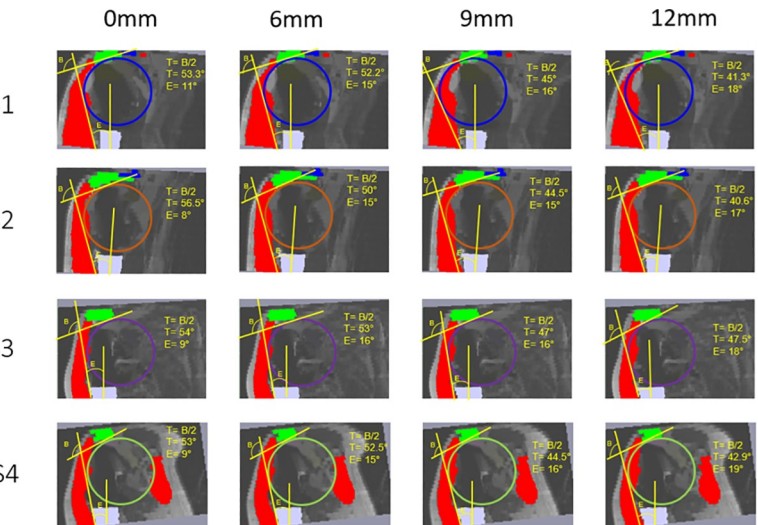

**Fig 7. Example of images extracted from SliceOmatic™ representing the different slices at each lateralization with angle measurements for Patient 1.** S1 = Slice 1; S2 = Slice 2; S3 = Slice 3; S4 = Slice 4; Blue Circle = 2D ILSS for Slice 1; Orange Circle = 2D ILSS for Slice 2; Purple Circle = 2D ILSS for Slice 3; Green Circle = 2D ILSS for Slice 4. In red: deltoid muscle; in green = acromion; in blue = clavicle; in white = humeral diaphysis. B = Angle B in degree; E = Angle E in degree; T = B/2.

when comparing lateralization results 2x2 (significance level was p> 0.05 for 6vs9, 6vs12, 9vs12). For DMA, there are no significant difference (significance level was p>0.05) when comparing initial position (0mm) versus 6mm, 9mm and 12mm; except for S1 and S2 when comparing 0mm versus 9mm and 12mm. (significance level was between p = 0.04* and p = 0.001*).

**Table 5. Comparison of the different lateralization simulations.**

|  | MD slice | *P* values | *P* values | *P* values | *P* values | *P* values | *P* values |
|---|---|---|---|---|---|---|---|
|  |  | **0vs6** | **0vs9** | **0vs12** | **6vs9** | **6vs12** | **9vs12** |
| Angle T | S1 | **0.002*** | **0.002*** | **0.002*** | 0.27 | **0.007*** | **0.05*** |
|  | S2 | **0.002*** | **0.002*** | **0.002*** | 0.19 | **0.006*** | **0.05*** |
|  | S3 | **0.002*** | **0.002*** | **0.002*** | 0.61 | 0.1 | 0.24 |
|  | S4 | **0.002*** | **0.002*** | **0.002*** | 0.48 | **0.05*** | 0.08 |
| Angle E | S1 | **0.04*** | **0.005*** | **0.002*** | 0.27 | **0.05*** | 0.22 |
|  | S2 | **0.01*** | **0.002*** | **0.002*** | 0.22 | 0.1 | 0.27 |
|  | S3 | **0.005*** | **0.002*** | **0.002*** | 0.14 | 0.1 | 0.5 |
|  | S4 | **0.02*** | **0.01*** | **0.002*** | 0.41 | 0.16 | 0.4 |
| **CER** | S1 | **0.0006*** | **0.002*** | **0.002*** | 0.56 | 0.06 | 0.22 |
|  | S2 | **0.0006*** | **0.002*** | **0.002*** | 0.27 | 0.11 | 0.22 |
|  | S3 | **0.002*** | **0.002*** | **0.002*** | 1.00 | 0.61 | 0.34 |
|  | S4 | **0.002*** | **0.002*** | **0.002*** | 0.75 | **0.05*** | 0.07 |
| **DMA** | S1 | 0.46 | **0.004*** | **0.001*** | **0.03*** | **0.01*** | 0.1 |
|  | S2 | 0.32 | **0.04*** | **0.03*** | 0.13 | **0.05*** | 0.48 |
|  | S3 | 0.38 | 0.21 | 0.1 | 0.46 | 0.16 | 0.54 |
|  | S4 | 0.46 | 0.26 | 0.1 | 0.54 | 0.38 | 0.54 |

* = significantly different.

## 4 Discussion

The aim of this study was to assess the effect of lateralizing the glenosphere on the DMA and coaptation role of MD. To this end, the reliability of the methodology needed to be evaluated. The rater's proficiency in identifying and selecting anatomical structures on MRI is key. The results of the inter- and intra-rater agreement assessment of the 3D models and ILSS positioning (Table 2) gave strength to our methods. They were also consistent with other studies describing the SliceOmatic™ software as a reliable 3D reconstruction tool [23, 24]. Therefore, using the model developed by Billuart *et al.* [20] and then by Hereter Gregori *et al.* [21], we compared four RSA lateralization options: 0, 6, 9, 12 mm. Our hypothesis that lateralization increases the coaptation role of the MD was confirmed. Lateralizing the glenoid implant led to a significant increase in the coaptation role of the Middle Deltoid. With a simulated lateralization of 6, 9 and 12 mm, the CER increased significantly compared to initial conditions (Table 4), with values closer to 1. This was especially true in the posterior (S1 = 0.90) and middle (S2 = 0.91) portions of MD. The glenoid implants are anatomically angled towards the S1 and S2 portions which may explain this trend. The CER seems to increase with lateralization but there was no significant difference when comparing lateralization results 2x2 (Table 5). Our findings are like those reported in the literature. Indeed, the studies published by Boileau *et al.* [10–12] or Hettrich *et al.* [4] and Hoenecke *et al.* [13], promote lateralization to improve mobility, stability and strength.

The two types of implants used in this study (*Onlay* for Wright-Tornier™ and *Inlay* for DePuy-Synthes™) do not have the same effect on lateralization. The Onlay is more lateralizing because of its design and its implantation concept. However, considering that each implant was compared to itself in the different lateralization simulations, the methodology remains sound.

Lateralization leads to tensioning of muscle fibers but its impact on DMA remains unclear. Werthel *et al.* [15, 16] showed that a few millimeters of lateralization did not affect the DMA. Our findings appear to agree with these results. Indeed, whatever the amount of lateralization, the DMA of the S3 and S4 portions were not significantly different, and 6 mm of lateralization had no effect on the DMA. However, a few millimeters' increase was seen in DMA with 9 mm and 12 mm of lateralization, but only for S1 and S2 (Tables 4 and 5). Moreover, lateralization did not appear to have a significant impact on DMA for all portions of the MD whereas there was a significant increase in the coaptation effect on the entire MD.

Clinically, the CER is a biomechanical factor that helps explain the Middle Deltoid's more prominent role in joint stability after RSA, as described in the literature [10–12]. This improvement in stability is therefore achieved by the Middle Deltoid actively, not only through passive fiber tensioning that results from humerus lengthening during surgery.

This study has limitations. The imaging is realized with a patient in supine position which could presses and deforms the posterior deltoid on the MRI. In addition, the supine position in the MRI scanner could change the position of the scapula on the rib cage. However, we positioned the arm so that it was not in extension. This positioning of the arm limits the modification of the anatomical relationships between the MD and the elements constituting the ILSS, inherent in the supine position. The methodology to identify the joint's center of rotation and the DMA was based on approximations. In this study, a choice was made to explore only the deltoid as a mechanical system, excluding other muscles (rotators'cuff, playing a key role on stability of the shoulder is not functional for these patients). Indeed, the MD has a key functional role after surgery since Li et al. [17] and Smith *al.* [18] showed a significant increase in the EMG activity of the MD after surgery during abduction. Furthermore, Yoon *et al.* [25] indicate that clinicians should pay attention to the deltoid volume for good functional

outcomes: a larger deltoid indicates a stronger muscle. Anatomically, MD has been specifically studied because of the 4 fibrous bands described by Sakoma *et al.* [22]. In our opinion, these fibrous bands reflect the adaptation of the MD to transmit forces to the underlying anatomical or prosthetic elements. The path of the anterior deltoid also reflecting on the ILSS may have a certain efficiency, but we did not assess this point quantitatively. The simplifying hypothesis developed to explain the lateralization of ILSS is considered acceptable. Indeed, the lateralization of the implant tends to increase the contacts between the ILSS and the MD and therefore to increase the coaptator effect of the MD. However, we have not quantitatively assessed this point. Finally, considering the lateral translation of the humerus with the lateralization of the glenosphere: translating the metaglene increases the coaptator effect of the MD. On the other hand, lateralization of the metaglene is accompanied by a lateralization of the humerus which therefore results in conservation of the DMA.

## 5 Conclusion

This study is the first to analyze the effects that lateralizing the glenosphere has on the coaptation role of the Middle Deltoid with a previously developed model. Simulating 6, 9 and 12 mm of lateralization on a personalized model derived from MRI showed improved coaptation of the Middle Deltoid, compared to no lateralization. The DMA was only affected starting at 9 mm of lateralization and only on some portions of the MD. The improvement in stability would then be the result of an increase in the coaptation role of the Middle Deltoid rather than an increase in its DMA. The coaptation effect of the MD could be doubly beneficial from a clinical standpoint. On the one hand, improved joint stability during anterior elevation would make it possible to lift heavier weights, and on the other, it would also help lower the dislocation rate.

## Acknowledgments

We thank Yves Martel for designing the new SliceOmatic™ tools and Helena Brunel for the statistical analysis. We are grateful to Joanne Archambault and Kathleen Beaumont for English language assistance.

## Author Contributions

**Conceptualization:** Lucas Martinez, Thomas Poirier, Fabien Billuart.

**Data curation:** Lucas Martinez, Margaux Machefert, Fabien Billuart.

**Formal analysis:** Lucas Martinez.

**Investigation:** Lucas Martinez, Margaux Machefert, Jean Matsoukis, Fabien Billuart.

**Methodology:** Lucas Martinez, Margaux Machefert, Thomas Poirier, Jean Matsoukis, Fabien Billuart.

**Project administration:** Lucas Martinez, Fabien Billuart.

**Resources:** Thomas Poirier, Jean Matsoukis.

**Software:** Lucas Martinez, Margaux Machefert.

**Supervision:** Lucas Martinez, Jean Matsoukis.

**Validation:** Lucas Martinez, Thomas Poirier, Jean Matsoukis, Fabien Billuart.

**Visualization:** Lucas Martinez, Thomas Poirier, Fabien Billuart.

**Writing – original draft:** Lucas Martinez, Margaux Machefert, Thomas Poirier, Jean Matsoukis, Fabien Billuart.

**Writing – review & editing:** Lucas Martinez, Margaux Machefert.

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
