## [Decision Letter · Decision Letter 0]

10 Jun 2021

PONE-D-21-12682

Analysis of the coaptation role of the deltoid in reverse shoulder arthroplasty. A preliminary biomechanical study.

PLOS ONE

Dear Dr. Martinez,

Thank you for submitting your manuscript to PLOS ONE. After careful consideration, we feel that it has merit but does not fully meet PLOS ONE’s publication criteria as it currently stands. Therefore, we invite you to submit a revised version of the manuscript that addresses the points raised during the review process.

We look forward to receiving your revised manuscript.

Kind regards,

Katherine Saul

Academic Editor

PLOS ONE

Journal Requirements:

2.Please provide additional details regarding participant consent. In the ethics statement in the Methods and online submission information, please ensure that you have specified (1) whether consent was informed and (2) what type you obtained (for instance, written or verbal, and if verbal, how it was documented and witnessed). If your study included minors, state whether you obtained consent from parents or guardians. If the need for consent was waived by the ethics committee, please include this information.

Reviewers' comments:

Reviewer's Responses to Questions

**Comments to the Author**

1. Is the manuscript technically sound, and do the data support the conclusions?

Reviewer #1: Yes

Reviewer #2: Yes

2. Has the statistical analysis been performed appropriately and rigorously? 

Reviewer #1: Yes

Reviewer #2: Yes

3. Have the authors made all data underlying the findings in their manuscript fully available?

Reviewer #1: Yes

Reviewer #2: Yes

4. Is the manuscript presented in an intelligible fashion and written in standard English?

Reviewer #1: Yes

Reviewer #2: Yes

5. Review Comments to the Author

Reviewer #1: Summary: This is a computer model simulation of deltoid coaptation with simulated increases in lateralization to the glenosphere based on MRI scans of patients that are postop from a non-lateralized rTSA. This study shows that lateralization does little to affect the moment arm of the deltoid, but does increase its coaptation. This gives credence that lateralization leads to increased stability in rTSA not through affecting the deltoid moment arm, but due to the coapt effect of the middle deltoid. Weaknesses to this study remain the relatively small number of patients included (n=7) and the variability of the implants used.

Line 43: Change main to common, it is less commonly seen with modern implant designs and improved techniques.

Line 218-222: Why was the second rater not used for patient’s 3-6? This should be addressed in the text

Reviewer #2: The authors attempt to model how lateralization of shoulder implant impacts the deltoid. It is an important and interesting topic, but it is a little difficult to for the reader to visualize and conceptualize. Including a set of images that contains all slices at all lateralizations with the angles overlay may help. There definitely should be a visualization for how the lateralization is modeled and how this affects the position and visualization of the deltoid.

Below are more specific comments:

Abstract:

Line 34: Should ‘portions’ be slices?

Introduction:

The introduction would benefit from a description of scapular notching, its causes, and its effects.

Line 61: I realize the CER has been described in previous literature, but since this is a major component of this study, I believe it needs to be described in the introduction. At least a basic definition.

Materials and Methods

Line 81: The results indicate that one participant was excluded due to movement during the MRI making the images unusable for the model creation. Here you imply that one subject was excluded based on exclusion criteria. Insufficient imaging is not an exclusion criteria.

Line 104/Fig 1: Identify what colors represent which anatomic segments.

Line 128: Is RCE the same as CER? If not, what is RCE?

Line 203: From the results, it looks like rater 2 only completed the model creation and not the ILSS positioning.

Results

Table 2: You may want to consider changing “model” to “week” in the last two rows of the table.

Lines 239-246: In the text I think you should list the overall (for all slices) mean and SD rather than the range. Additionally you should list the overall mean and SD for the DMA

Line 254: Typically significance levels are presented as p=0.002, etc.

Discussion

Line 283: Can you explain the different effects on lateralization. Did you see differences in the two patients with the Onlay for Wright-Tornier implants?

6. PLOS authors have the option to publish the peer review history of their article (what does this mean?). If published, this will include your full peer review and any attached files.

Reviewer #1: No

Reviewer #2: No

---

## [Author Response · Author response to Decision Letter 0]

30 Jun 2021

To the Editorial Board and the Reviewers,

First, we would like to thank you deeply for all the precious comments the 2 reviewers have done for our manuscript. These comments have helped us to improve the quality and the robustness of our manuscript. We have meticulously considered all the comments made by the 2 reviewers. We are very honored to submit our new manuscript and we hope that this new version will meet all your expectations. Please find below our answers to each point mentioned by the 2 reviewers. As requested, you will find in our new submission, a version highlighting the changes (Revised Manuscript with tracked changes) made with comments visible in Word with elements highlighted in yellow and an unmarked version without tracked changes. Please note that we have chosen to add a new author to the manuscript (Margaux Machefert, PT). Indeed, her contribution was very important during the review of our manuscript. During the first submission, she did not have time to complete her work for the study. She was able to do so for the reviewing.

For Reviewer #1:

We thank you very much for this very insightful review of our manuscript. In response to your remark about the weakness of the study relating to the small sample: yes, the sample of this preliminary study is small, but we think that our results are reliable given that the values all evolve in the same direction for all the subjects. More about implants, the 2 implants do not have the same effect on lateralization. The Onlays are more lateralizing. However, we do not compare the Onlay / Inlay implants in this study but rather different simulations of lateralizations for each patient.

Line 43 : Change main to common, it is less commonly seen with modern implant designs and improved techniques.

We replaced the word "main" by the word “common", according to your advice.

Line 218-222 : Why was the second rater not used for patient’s 3-6? This should be addressed in the text.

Rater 2 was a PT student, she participated in the reliability study for this project, constituting her thesis for PT Diploma. At the time we decided to submit our work, she had not finished carrying out all the 3D modeling, ILSS positioning and the statistical part. We made the choice to publish our work quickly. Since then, she has been able to complete the reliability study and we have been able to add her data in this new version.

For Reviewer #2:

We thank you for your careful review of our work and your comments which have greatly improved our manuscript. We took the time to take each of your comments into consideration. Your main remark concerned the addition of figures to improve the understanding of our methodology. We have thus added 3 new Figures that will improve these points. We hope that these figures will meet your expectations

The introduction would benefit from a description of scapular notching, its causes, and its effects.

We added a description of notching in the introduction (its causes and effects) in accordance with your remark.

Line 34: Should ‘portions’ be slices?

You are right, it is indeed "slice" and not "portion". Even if in French it is the same word.

Line 61: I realize the CER has been described in previous literature, but since this is a major component of this study, I believe it needs to be described in the introduction. At least a basic definition.

We added a basic description of CER to make the study easier to understand.

Line 81: The results indicate that one participant was excluded due to movement during the MRI making the images unusable for the model creation. Here you imply that one subject was excluded based on exclusion criteria. Insufficient imaging is not an exclusion criteria.

We forgot to mention that movements during the acquisition of the MRI images (making the images uninterpretable), was a criterion for excluding the patient. We apologize for this oversight. One of the patients was therefore effectively excluded after the MRI examination, because images could not be interpreted.

Line 104/Fig 1: Identify what colors represent which anatomic segments.

We update the caption with the colors representing the anatomic segments. 

Line 128: Is RCE the same as CER? If not, what is RCE?

It is indeed the CER! It is an oversight of translation. in French CER is written RCE for “Ratio Coaptateur/Elévateur”. We apologize for this mistake.

Line 203: From the results, it looks like rater 2 only completed the model creation and not the ILSS positioning. 

We echo the remark also made by reviewer 1: Rater 2 was a PT student, she participated in the reliability study for this project, constituting her thesis for PT Diploma. At the time we decided to submit our work, she had not finished carrying out all the 3D modeling, ILSS positioning and the statistical part. We made the choice to publish our work quickly. Since then, she has been able to complete the reliability study and we have been able to add her data in this new version.

Table 2: You may want to consider changing “model” to “week” in the last two rows of the table.

We replaced the term "model" with "week" as requested.

Lines 239-246: In the text I think you should list the overall (for all slices) mean and SD rather than the range. Additionally you should list the overall mean and SD for the DMA

As requested, we have included the mean and standard deviation of each parameter for each lateralization and each slice, directly in the text of Results, part “Biomechanical parameters”.

Line 254: Typically significance levels are presented as p=0.002, etc.

As requested, we added "p =" before the significance levels for this paragraph and wherever needed in the manuscript.

Line 283: Can you explain the different effects on lateralization. Did you see differences in the two patients with the Onlay for Wright-Tornier implants? 

In response to this remark, indeed the 2 implants do not have the same effect on lateralization. The onlays are more lateralizing. However, we do not compare the onlay / inlay in this study but rather different simulations of lateralizations for each patient.

We thank you in advance for the new reviewing and we hope that you will find it suitable for your journal. Do not hesitate to contact us with any questions (l.martinez@ifmk.fr; + 33 684174961).

Yours truly,

---

## [Decision Letter · Decision Letter 1]

21 Jul 2021

PONE-D-21-12682R1

Analysis of the coaptation role of the deltoid in reverse shoulder arthroplasty. A preliminary biomechanical study.

PLOS ONE

Dear Dr. Martinez,

Thank you for submitting your manuscript to PLOS ONE. After careful consideration, we feel that it has merit but does not fully meet PLOS ONE’s publication criteria as it currently stands. Therefore, we invite you to submit a revised version of the manuscript that addresses the points raised during the review process.

We look forward to receiving your revised manuscript.

Kind regards,

Katherine Saul

Academic Editor

PLOS ONE

Journal Requirements:

Reviewers' comments:

Reviewer's Responses to Questions

**Comments to the Author**

1. If the authors have adequately addressed your comments raised in a previous round of review and you feel that this manuscript is now acceptable for publication, you may indicate that here to bypass the “Comments to the Author” section, enter your conflict of interest statement in the “Confidential to Editor” section, and submit your "Accept" recommendation.

Reviewer #2: All comments have been addressed

2. Is the manuscript technically sound, and do the data support the conclusions?

Reviewer #2: Yes

3. Has the statistical analysis been performed appropriately and rigorously? 

Reviewer #2: Yes

4. Have the authors made all data underlying the findings in their manuscript fully available?

Reviewer #2: Yes

5. Is the manuscript presented in an intelligible fashion and written in standard English?

Reviewer #2: Yes

6. Review Comments to the Author

Reviewer #2: The authors did a commendable job addressing all comments and concerns.

I apologize for any confusion. My previous comment "Lines 239-246: In the text I think you should list the overall (for all slices) mean and SD rather than the range. Additionally you should list the overall mean and SD for the DMA" I was suggesting the average for all slices, not the average of each slice. Your edits essentially repeat the data from the table in the text. I believe the previous version had one range for slices 1 through 4, I was simply suggesting replace the range with one mean of slices 1 through 4.

7. PLOS authors have the option to publish the peer review history of their article (what does this mean?). If published, this will include your full peer review and any attached files.

Reviewer #2: No

---

## [Author Response · Author response to Decision Letter 1]

23 Jul 2021

To the Editorial Board and the Reviewers,

We would like to thank again the Editorial Board and the Reviewers for the new precious comments they have done for our manuscript. These last comments will certainly make the manuscript more robust and understandable. We have considered the comment of Reviewer 2 concerning the Results and we have modified accordingly. We apologize for our misunderstanding of his first remark. We hope that our modifications will correspond to his expectations. Likewise, in accordance with your request, we have added information on patients' consent to participate in the study and how consent was obtained. Again, we are very honored to submit our new manuscript and we hope that this new version will meet all your expectations. We left in this letter the comments of the first review from the 2 reviewers. We have also highlighted the comment of this new reviewing with our response. As requested, you will find in our new submission, a version highlighting the change made with comments visible in Word with elements highlighted in yellow (Revised Manuscript with tracked changes). Please note that only the last comment from Reviewer 2 appears in this version. You will also find an unmarked version without tracked changes. 

First reviewing

For Reviewer #1:

We thank you very much for this very insightful review of our manuscript. In response to your remark about the weakness of the study relating to the small sample: yes, the sample of this preliminary study is small, but we think that our results are reliable given that the values all evolve in the same direction for all the subjects. More about implants, the 2 implants do not have the same effect on lateralization. The Onlays are more lateralizing. However, we do not compare the Onlay / Inlay implants in this study but rather different simulations of lateralizations for each patient.

Line 43 : Change main to common, it is less commonly seen with modern implant designs and improved techniques.

We replaced the word "main" by the word “common", according to your advice.

Line 218-222 : Why was the second rater not used for patient’s 3-6? This should be addressed in the text.

Rater 2 was a PT student, she participated in the reliability study for this project, constituting her thesis for PT Diploma. At the time we decided to submit our work, she had not finished carrying out all the 3D modeling, ILSS positioning and the statistical part. We made the choice to publish our work quickly. Since then, she has been able to complete the reliability study and we have been able to add her data in this new version.

For Reviewer #2:

We thank you for your careful review of our work and your comments which have greatly improved our manuscript. We took the time to take each of your comments into consideration. Your main remark concerned the addition of figures to improve the understanding of our methodology. We have thus added 3 new Figures that will improve these points. We hope that these figures will meet your expectations

The introduction would benefit from a description of scapular notching, its causes, and its effects.

We added a description of notching in the introduction (its causes and effects) in accordance with your remark.

Line 34: Should ‘portions’ be slices?

You are right, it is indeed "slice" and not "portion". Even if in French it is the same word.

Line 61: I realize the CER has been described in previous literature, but since this is a major component of this study, I believe it needs to be described in the introduction. At least a basic definition.

We added a basic description of CER to make the study easier to understand.

Line 81: The results indicate that one participant was excluded due to movement during the MRI making the images unusable for the model creation. Here you imply that one subject was excluded based on exclusion criteria. Insufficient imaging is not an exclusion criteria.

We forgot to mention that movements during the acquisition of the MRI images (making the images uninterpretable), was a criterion for excluding the patient. We apologize for this oversight. One of the patients was therefore effectively excluded after the MRI examination, because images could not be interpreted.

Line 104/Fig 1: Identify what colors represent which anatomic segments.

We update the caption with the colors representing the anatomic segments. 

Line 128: Is RCE the same as CER? If not, what is RCE?

It is indeed the CER! It is an oversight of translation. in French CER is written RCE for “Ratio Coaptateur/Elévateur”. We apologize for this mistake.

Line 203: From the results, it looks like rater 2 only completed the model creation and not the ILSS positioning. 

We echo the remark also made by reviewer 1: Rater 2 was a PT student, she participated in the reliability study for this project, constituting her thesis for PT Diploma. At the time we decided to submit our work, she had not finished carrying out all the 3D modeling, ILSS positioning and the statistical part. We made the choice to publish our work quickly. Since then, she has been able to complete the reliability study and we have been able to add her data in this new version.

Table 2: You may want to consider changing “model” to “week” in the last two rows of the table.

We replaced the term "model" with "week" as requested.

Lines 239-246: In the text I think you should list the overall (for all slices) mean and SD rather than the range. Additionally you should list the overall mean and SD for the DMA

As requested, we have included the mean and standard deviation of each parameter for each lateralization and each slice, directly in the text of Results, part “Biomechanical parameters”.

Line 254: Typically significance levels are presented as p=0.002, etc.

As requested, we added "p =" before the significance levels for this paragraph and wherever needed in the manuscript.

Line 283: Can you explain the different effects on lateralization. Did you see differences in the two patients with the Onlay for Wright-Tornier implants? 

In response to this remark, indeed the 2 implants do not have the same effect on lateralization. The onlays are more lateralizing. However, we do not compare the onlay / inlay in this study but rather different simulations of lateralizations for each patient.

Second reviewing

“I apologize for any confusion. My previous comment "Lines 239-246: In the text I think you should list the overall (for all slices) mean and SD rather than the range. Additionally you should list the overall mean and SD for the DMA" I was suggesting the average for all slices, not the average of each slice. Your edits essentially repeat the data from the table in the text. I believe the previous version had one range for slices 1 through 4, I was simply suggesting replace the range with one mean of slices 1 through 4.”

We apologize for misunderstanding the request. As requested by Reviewer 2, we have re-modified the results of “Biomechanical Parameters” as follows: we have computed one Mean±SD of slices 1 through 4 for each parameters and each lateralization (0, 6, 9, 12mm) : Angle T, E, CER and DMA. In this way, we avoid repeating the Table 4. We hope this will meet your expectations.

Likewise, in accordance with your request, we have added information on patients' consent to participate in the study and how consent was obtained: 

Line 86-88: Each patient was informed of the study via a newsletter. Then, each patient gave their consent to participate in the study by signing a consent letter informing them that their data will be used for the current study. Patient data has been anonymized.

We thank you in advance for the new reviewing and we hope that you will find it suitable for your journal. Do not hesitate to contact us with any questions (l.martinez@ifmk.fr; + 33 684174961).

Yours truly,

---

## [Editor Report · Decision Letter 2]

26 Jul 2021

Analysis of the coaptation role of the deltoid in reverse shoulder arthroplasty. A preliminary biomechanical study.

PONE-D-21-12682R2

Dear Dr. Martinez,

We’re pleased to inform you that your manuscript has been judged scientifically suitable for publication and will be formally accepted for publication once it meets all outstanding technical requirements.

Kind regards,

Katherine Saul

Academic Editor

PLOS ONE
---

## [Editor Report · Acceptance letter]

5 Aug 2021

PONE-D-21-12682R2 

Analysis of the coaptation role of the deltoid in reverse shoulder arthroplasty. A preliminary biomechanical study. 

Dear Dr. Martinez:

I'm pleased to inform you that your manuscript has been deemed suitable for publication in PLOS ONE. Congratulations! Your manuscript is now with our production department. 

Kind regards, 

on behalf of

Dr. Katherine Saul 

Academic Editor

PLOS ONE